# Temporal Understanding of Gaze Communication with GazeTransformer

**Ryan Anthony de Belen**                                    R.DEBELEN@UNSW.EDU.AU
**Gelareh Mohammadi**                                        G.MOHAMMADI@UNSW.EDU.AU
**Arcot Sowmya**                                             A.SOWMYA@UNSW.EDU.AU
*University of New South Wales, Sydney, NSW, Australia*

## Abstract

Gaze plays a crucial role in daily social interactions as it allows humans to communicate intentions effectively. We address the problem of temporal understanding of gaze communication in social videos in two stages. First, we develop GazeTransformer, an end-to-end module that infers atomic-level behaviours in a given frame. Second, we develop a temporal module that predicts event-level behaviours in a video using the inferred atomic-level behaviours. Compared to existing methods, GazeTransformer does not require human head and object locations as input. Instead, it identifies these locations in a parallel and end-to-end manner. In addition, it can predict the attended targets of all predicted humans and infer more atomic-level behaviours that cannot be handled simultaneously by previous approaches. We achieve promising performance on both atomic- and event-level prediction on the (M)VACATION dataset. Code will be available at https://github.com/gazetransformer/gazetransformer.

**Keywords:** Gaze estimation and prediction, gaze communication behaviour prediction

## 1. Introduction

Apart from enriching verbal utterances, non-verbal communication in itself plays an important role in conveying important information (Mehrabian, 2017). In addition, the ability to learn and understand gaze communication plays a crucial role in the development of social cognition, information processing and language (Brooks and Meltzoff, 2015; Adamson et al., 2004; de Belen et al., 2020, 2023a, 2021, 2023b). In the first years of life, infants learn to follow gaze and coordinate their attention with that of their primary caregivers (Brooks and Meltzoff, 2015). For example, an infant may respond by following the gaze point of the parent looking at a target object. Difficulties in understanding gaze communication may result in various socio-communicative impairments during development, as the former makes it challenging to associate a word with an object (Mundy, 2018). This motivates researchers to systematically study gaze and responses to this primitive form of communication.

Although not the primary focus of this paper, it is worth mentioning that the mature saliency estimation domain (Borji, 2019; de Belen et al., 2022) bears similarity to gaze behaviour prediction. With distinct differences, both aim to computationally model the mechanisms that underlie human visual attention. Instead of inferring a person's attended target in an image, saliency estimation aims to identify the pixel locations that can attract the attention of humans while viewing images. In fact, a salient region in an image will most likely attract the attention of a person in the image. This is why most previous works on human gaze behaviour prediction contain a scene branch used for saliency estimation.

Earlier works have demonstrated the ability of neural networks to estimate the attended visual target of a person in an image (Recasens et al., 2015). The most common architecture includes two branches: a head branch that learns head pose features and a scene branch that learns salient regions in the image. Succeeding works have extended the problem to handling out-of-frame gaze targets, resulting in better performance (Chong et al., 2020, 2018). Furthermore, recent works have explored the problem of estimating attended visual targets in 360°images (Li et al., 2021), video (Recasens et al., 2017; Fang et al., 2021) and even 3D space (Massé et al., 2017; Wei et al., 2018; Brau et al., 2018; Massé et al., 2019; Hu et al., 2022a). These approaches are also useful for understanding human gaze communication behaviours in social video.

Over the past decades, different frameworks have been proposed in psychology and neuroscience to study and understand gaze communication (Itier and Batty, 2009; Jording et al., 2018). A recent framework involves breaking down gaze behaviours into its atomic- and event-level components (Fan et al., 2019). Atomic-level components include the following fine-grained gaze patterns: (1) *Single* is the simplest and does not involve any social communication/interaction behaviour. (2) *Mutual* occurs when two people look at each other. (3) *Avert* happens when a person looks away from another's gaze. (4) *Refer* occurs when a person tries to direct the attention of another person to an object. (5) *Follow* happens when another person responds to an initiation of attention of another person to an object. (6) *Share* occurs when two people are looking at the same object. On the other hand, gaze communication events are coarse-grained and include the following: (1) *Non-communicative* (2) *Mutual Gaze* (3) *Gaze Aversion* (4) *Gaze Following* and (5) *Joint Attention*. These events can be formed by temporally combining atomic-level components. In this work, we adopt a slightly different framework and provide justifications for this minor change in Section 3.

Previous works have explored the capability of deep learning networks to predict gaze communication of people in images or videos (Fan et al., 2019; Guo et al., 2022; Marin-Jimenez et al., 2019; Marín-Jiménez et al., 2021). Most approaches require human and object bounding boxes or use a decoupled head detector (Marin-Jimenez et al., 2019; Fan et al., 2019; Marín-Jiménez et al., 2021), resulting in sub-optimal solutions. A better solution is to use an end-to-end module that jointly learns to predict head and object bounding boxes and their corresponding gaze relationships. Recently, an end-to-end model has been proposed to estimate the location of the attended target (Tu et al., 2022), while another end-to-end model can only detect *Mutual* gaze relationships (Guo et al., 2022), limiting future applications for handling more complex gaze communication. In contrast, our proposed atomic-level module can predict *Single*, *Mutual* and *Share* gaze relationships, while our temporal module can predict event-level gaze communication.

Our framework to predict human gaze communication behaviours consists of two stages:

1. We develop GazeTransformer, an end-to-end module that identifies atomic-level gaze communication. The module consists of an image feature extractor backbone, a transformer encoder-decoder network and several multi-layer perceptrons that predict human head and object locations and their corresponding gaze relationships (i.e., human-target interaction (HTI) instances), in parallel. These HTI instances are elements of an adjacency matrix that we use for inferring atomic-level gaze communication. Gaze-Transformer achieves promising results on the (M)VACATION dataset.

2. We develop a temporal module for predicting event-level gaze communication behaviours. The model architecture consists of a long short-term memory network and a fully connected layer for classification. It first processes all video frames and determines unique atomic labels within the video. As a result, it does not miss crucial frames which will otherwise be excluded if a widely-used uniform sub-sampling approach (Carreira and Zisserman, 2017) is adopted. This module achieves promising results on the (M)VACATION dataset.

This paper makes the following contributions:

1. We modify the VACATION dataset and include atomic-level components that are minimally sufficient to build event-level gaze events. The reasons for this change are discussed in Section 3. These modifications are justified and result in a simpler, more efficient and effective training and evaluation paradigm. Furthermore, the new atomic-level labels provide a more practical categorisation since they do not require temporal information for inference. This is ideal because atomic-level labels are defined for each frame instead of being reliant on previous or succeeding frames.

2. We present GazeTransformer, a novel end-to-end model that predicts human head and object locations in parallel. In addition, it predicts the attended targets of all the predicted humans in the scene. Furthermore, we propose a novel way to infer atomic-level labels (e.g., *Single*, *Mutual*, *Share*) from adjacency matrices. Currently, this ability cannot be handled simultaneously by existing end-to-end models.

3. We present a temporal model for predicting event-level gaze communication behaviours from the atomic-level gaze communication already detected.

## 2. Related Work

The ability to use and understand gaze allows humans to communicate and share intentions effectively. In addition, it provides a means for evaluating another person's interest in the environment (Mehrabian, 2017). The white part of the eye, called the sclera, is more prominent in humans than in other mammals, allowing humans to leverage the colour difference between the sclera and the darker-coloured iris when directing their attention to a potential target for conveying intention (Kobayashi and Kohshima, 1997). When the pupil information cannot be reliably used for communication, humans resort to using head orientation as another way to convey and infer intentions. Finally, if the eyes and head are occluded, body orientation provides a sufficient cue for communication. As a result, the ability to estimate gaze and head direction is crucial for humans in determining gaze communication in an image or video. Consider an image where two people are looking at an object (i.e., sharing attention). A person looking at this image needs to have the ability to determine the attended targets of both persons and understand that the targets are the same object. It is therefore appropriate to discuss related work under the following headings: gaze target prediction and gaze communication behaviour understanding.

**Gaze target prediction** Previous works focus on detecting gaze targets in an image (Recasens et al., 2015; Chong et al., 2018, 2020; Tu et al., 2022; Guan et al., 2020; Lian et al., 2018; Zhao et al., 2020; Bao et al., 2022; Hu et al., 2022b; Gupta et al., 2022), 360°image

(Li et al., 2021), video (Recasens et al., 2017; Fang et al., 2021), or 3D space (Massé et al., 2017; Wei et al., 2018; Brau et al., 2018; Massé et al., 2019; Hu et al., 2022a). Since our work focusses on images and videos, we review related works in more detail.

The earliest deep learning model follows a two-branch network approach that involves a head pathway and a scene pathway to infer a heatmap of the attended target of a person in an image (Recasens et al., 2015). An extension to this work involved considering body pose (Guan et al., 2020) and learning a modulation constant to identify out-of-frame targets in images, as well as in videos (Chong et al., 2018, 2020), resulting in improved prediction performance. Another two-stage method that aims to draw sight lines and determine the attended target by stopping at a position with high sight line strength has been proposed (Zhao et al., 2020). A three-stage method that uses depth and 3D gaze estimation was proposed to exclude predictions that are at improper depth (Fang et al., 2021). Another similar framework infers 3D geometry from a 2D image and parses the scene to infer the target gaze position in 2D (Bao et al., 2022). Another three-stage method uses a head branch, scene branch and a relational branch to identify the attended target (Chen et al., 2021; Hu et al., 2022b). A modular multimodal model leverages depth and pose estimation and can be used in privacy-sensitive settings (Gupta et al., 2022). A recent work presents a unified framework for jointly solving gaze estimation, gaze target prediction and gaze target detection (Wang et al., 2022). As can be observed from the above discussion, most prior approaches require ground truth locations of the human heads for accurate prediction of the attended targets, limiting their adoption to practical applications. A recent work addresses this issue using an end-to-end model that can simultaneously predict head bounding boxes and their corresponding gaze targets (Tu et al., 2022), while another determines *Mutual* gaze relationships (Guo et al., 2022). In contrast, we propose an end-to-end solution for simultaneously predicting the attended targets of each detected person in the scene, as well as inferring the corresponding *Single*, *Mutual* and *Share* gaze relationships in this work.

**Gaze communication behaviour understanding** Previous works detect if two persons are looking at each other (Marín-Jiménez et al., 2011; Marin-Jimenez et al., 2014, 2019; Marín-Jiménez et al., 2021; Palmero et al., 2018; Doosti et al., 2021; Guo et al., 2022), determine if two or more persons are sharing attention (Sumer et al., 2020), predict the common gaze target of a group of persons (Fan et al., 2018; Zhuang et al., 2019), or recognise atomic-level (e.g., *Single*, *Mutual*, *Share*) and event-level gaze communication behaviours (e.g., *Gaze Aversion*, *Joint Attention*) (Fan et al., 2019). Similar to the prior works on gaze target prediction, most approaches in this domain require ground truth locations of the humans in the scene.

In this work, we present GazeTransformer, an end-to-end module for atomic-level prediction. Compared to a previous model (Fan et al., 2019) that requires human and object locations, GazeTransformer automatically predicts these locations, the attended targets, and their corresponding gaze relationships in parallel. Unlike previous end-to-end models that can only identify attended targets (Tu et al., 2022) and handle *Mutual* gaze (Guo et al., 2022), GazeTransformer can simultaneously infer *Single*, *Mutual* and *Share* gaze relationships. A temporal module is also built on top of GazeTransformer for event-level classification. Our experimental results show that our atomic- and event-level modules achieve promising performance on the (M)VACATION dataset, which is a modified VACATION dataset described in Section 3.

|  | Number of instances | | | | | |
|---|---|---|---|---|---|---|
|  | *Single* | *Mutual* | *Avert* | *Refer* | *Follow* | *Share* |
| VACATION | 80,370 | 37,441 | 9,333 | 2,549 | 3,429 | 28,821 |
| (M)VACATION | 87,998 | 40,497 | - | - | - | 33,452 |

Table 1: Comparison of atomic-level gaze communication labels between VACATION and (M)VACATION datasets.

## 3. (M)VACATION dataset

The Video gAze CommunicATION (VACATION) dataset (Fan et al., 2019) is a large-scale video dataset that aims to tackle the problem of understanding human gaze communication in social videos from both atomic- and event- levels. It contains 300 videos of diverse social scenes with complete annotations of the bounding box locations of objects and human faces, human attention, and both atomic- and event-level gaze communication labels.

In the VACATION dataset, atomic-level labels were categorised into six classes: *Single, Mutual, Avert, Refer, Follow* and *Share*. On the other hand, event-level labels were composed of *Non-communicative, Mutual Gaze, Gaze Aversion, Gaze Following* and *Joint Attention*. While atomic-level labels are provided for each person in each frame, event-level labels are the same for an entire video/segment. Note that there is an imbalance in the number of instances for *Avert, Refer* and *Follow* atomic-level labels in the original VACATION dataset, as shown in Table 1.

As can be observed, there are more *Single* (i.e., no gaze interaction between persons in the scene), *Mutual* (i.e., two persons are looking at each other) and *Share* (i.e., two or more persons are looking at the same object) atomic-level gaze communication behaviours. On the other hand, there is substantially less number of *Avert* (i.e., one person looks away after another person gazes), *Refer* (i.e., one person tries to refer another person to another object by a mutual gaze followed by a look at an object) and *Follow* (i.e., a person looks at where another person is looking at) behaviours in the VACATION dataset.

While the VACATION dataset undeniably provides a useful baseline to develop computational models for gaze communication behaviour understanding, we believe that it requires minor changes for an easier, more efficient and effective training and evaluation paradigm. Therefore, we introduce a modified VACATION dataset, named (M)VACATION. The differences between the original and the modified datasets and the reasons for the modifications are outlined below:

1. the number of atomic-level gaze communication labels has been reduced to three: *Single, Mutual* and *Share*. We believe that these three fine-grained components are sufficient to build more complex and course-grained event-level gaze communication. In fact, the removed atomic-level labels (e.g., *Avert, Refer* and *Follow*) require temporal information, defeating their definition as atomic components. To illustrate, consider a *Joint Attention* scenario in Figure 1 where *Person1* shares attention to an object with *Person2*. In row 1, the ground truth atomic-level labels for *Person 1* is *Follow* while it is *Single* then *Refer* for *Person2* across several frames. However, it is

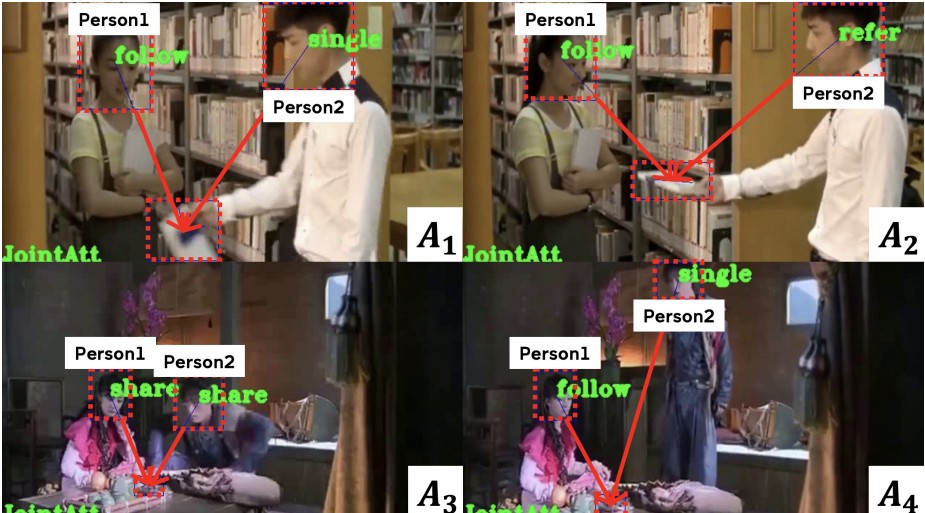

Figure 1: Frames with ground truth human head/object locations with their corresponding atomic-level labels from the VACATION dataset. As shown by the directed arrows in each frame, all persons are looking at the same object. However, the ground truth may either be: *Follow*, *Single*, *Refer*, or *Share*. Since atomic-level labels are defined per frame, we believe that the mentioned labels should be *Share*.

difficult even for a human to determine these defined ground-truth atomic-level labels for each frame without looking at the surrounding frames. A similar issue can be found in row 2 of Figure 1 in which *Person1* has a *Share* then *Follow* labels while *Person2* has *Share* then *Single* labels. Since the prediction of atomic-level labels is performed for each frame, *Single*, *Mutual* and *Share* provide a more practical categorisation of atomic-level labels in this case.

2. the atomic-level ground truth has been modified to ensure labelling consistency. This was easily automated by constructing adjacency matrices (defined and discussed more thoroughly in section 4.2) that denote directed gaze in a scene. Afterwards, the atomic-level labels are inferred from the adjacency matrices. As shown in Figure 1, while consecutive frames show that the two persons *Share* attention, the ground truth labels are different, making the problem unnecessarily complex. The underlying adjacency matrices, as will be discussed in section 4.2, are the same:

$$A_1 = A_2 = A_3 = A_4 = \begin{bmatrix} 0 & 0 & 0 \\ 1 & 0 & 0 \\ 1 & 0 & 0 \end{bmatrix}$$

3. After the proposed modifications, the (M)VACATION has a more balanced number of classes compared to the original VACATION dataset, as shown in Table 1. In addition to the advantages described above, training a deep learning model on a more balanced dataset results in better performance, especially at times when it is difficult to obtain representative examples in each class.

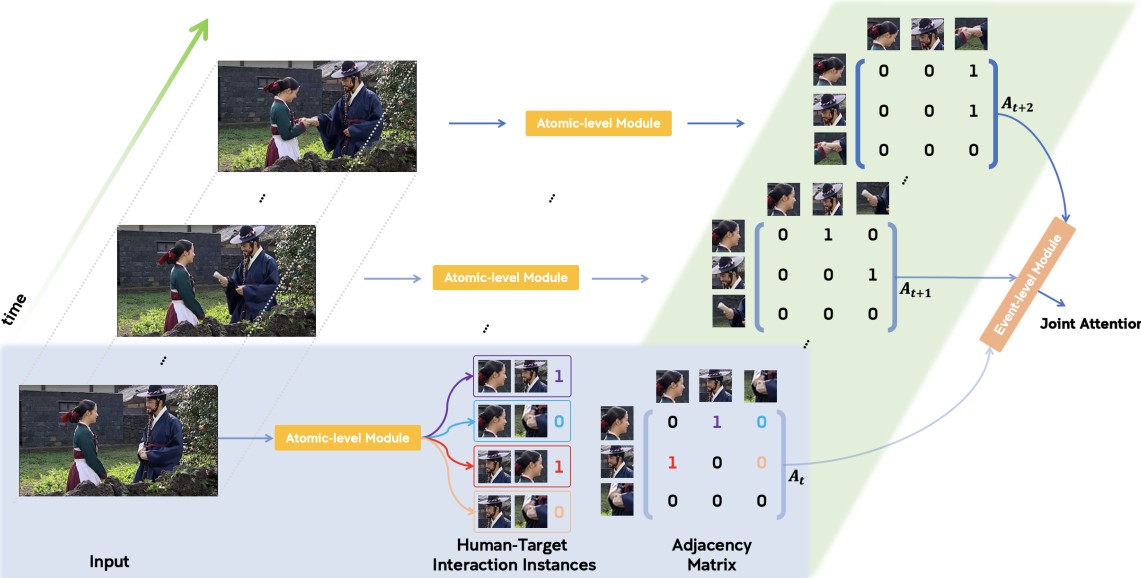

Figure 2: Overview of the proposed pipeline that consists of a two-step approach: the atomic-level module predicts the atomic-level behaviours of all persons in a given frame in parallel and end-to-end. Afterwards, the event-level module infers the event-level behaviour using the unique atomic-level prediction in a given video.

## 4. Methodology

As shown in Figure 2, an event, such as *Joint Attention*, consists of a series of temporally changing human-human or human-object interactions, which we call human-target interaction (HTI) instances. Clearly, it is important to recognise these atomic components first before attempting to understand more complex gaze communication. Therefore, we develop a two-step approach for the temporal understanding of gaze communication, as illustrated in Figure 2:

1. **Atomic-level prediction:** we develop GazeTransformer that predicts human/object locations, gaze targets and all HTI instances in a parallel and end-to-end manner. These HTI instances are elements of an adjacency matrix that we then use to infer the atomic labels of each person in the scene.

2. **Event-level prediction:** we develop a temporal module on top of GazeTransformer to predict the event labels using the unique atomic predictions in a video.

These two modules are trained separately to address the atomic- and event-level predictions. Afterwards, the output of the atomic-level module is passed to the event-level module. While our approach looks similar to Fan et al. (2019), we neither follow a neural message passing framework of graph neural networks nor require human head/object locations to generate the atomic labels. For event-level prediction, our approach uses the predicted adjacency matrix, while Fan et al. (2019) uses atomic-level transition and frequency counts as input.

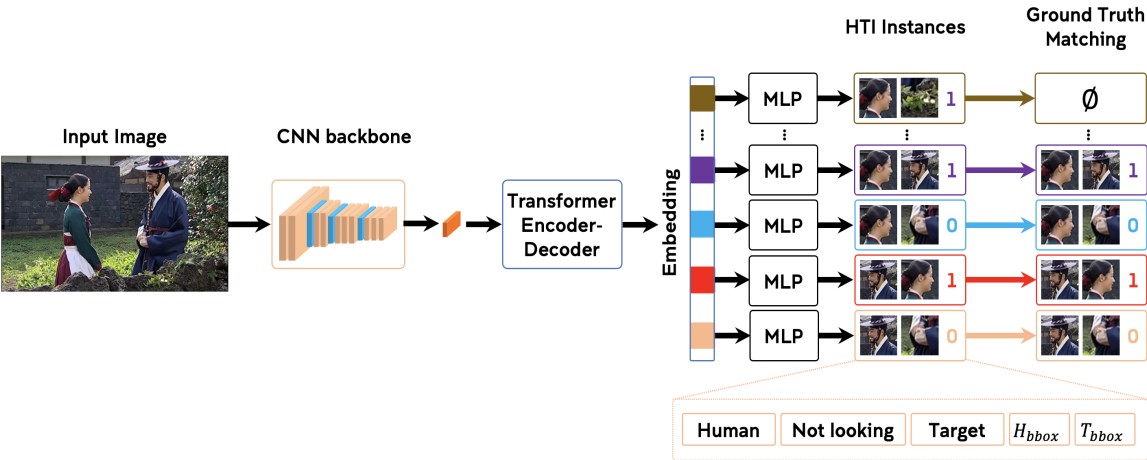

Figure 3: The proposed architecture of the atomic-level prediction module. It consists of three components: (1) an image feature extractor backbone (2) a transformer encoder-decoder network and (3) several multi-layer perceptrons to predict the HTI instances of a frame, in parallel. These HTI instances are elements of an adjacency matrix that we use to infer the atomic-level gaze communication behaviours. This resembles a set prediction problem.

### 4.1. Problem Formulation

Similar to the problem of learning human-object interactions (Qi et al., 2018), gaze communication behaviour prediction can be solved by analysing complete scene graphs (i.e., social graphs) and generating a sub-graph that entails the true gaze communication behaviours of persons in the scene.

A complete social graph is represented as $\mathcal{G} = (\mathcal{V}, \mathcal{E})$. Nodes $v \in \mathcal{V}$ take unique values from $\{1, ..., |\mathcal{V}|\}$ and represent distinct entities (e.g., human, object) in the scene. Edges $e \in \mathcal{E}$ are two-tuples $e = (v, w) \in \mathcal{V} \times \mathcal{V}$ and represent directed edges $v \to w$ that show all the possible human-human gaze interactions or human-object gaze interactions (i.e., HTI instances). The sub-graph $g = (\mathcal{V}_g, \mathcal{E}_g)$, where $\mathcal{V}_g \subseteq \mathcal{V}$ and $\mathcal{E}_g \subseteq \mathcal{E}$, denotes the true HTI instances in the scene. This $g$ is represented as an adjacency matrix $A = [0,1]^{|\mathcal{V}| \times |\mathcal{V}|}$. While this scene representation is similar to Fan et al. (2019), there are significant differences. Unlike (Fan et al., 2019), we consider the problem of finding the sub-graph $g$ as a set prediction problem. More specifically, the off-diagonals of $A$ are the HTI instances that the GazeTransformer predicts in a parallel and end-to-end way. Furthermore, we set the diagonals of $A$ to zero since we assume that (human) nodes do not look at themselves. In addition, we neither add a dummy node to represent the social scene nor set the maximum number of nodes for atomic classification. We also do not learn an additional readout function that is part of a graph neural network to generate the atomic labels. Instead, we use a simple, effective and practical way to infer atomic labels from an adjacency matrix (see Section 4.2).

## 4.2. Atomic-level prediction module

The atomic-level prediction module, GazeTransformer, consists of three components: (1) an image feature extractor backbone (2) a transformer encoder-decoder network and (3) several multi-layer perceptrons (MLPs). Unlike a previous method (Fan et al., 2019), we solve the problem of atomic-level prediction in an end-to-end manner. Unlike a previous end-to-end model (Guo et al., 2022) that can only handle *Mutual* gaze, GazeTransformer can infer *Single*, *Mutual* and *Share* gaze. As shown in Figure 3, it takes in an image frame and outputs all HTI instances in parallel (similar to a set prediction problem), effectively generating the elements of the adjacency matrix $A$ of sub-graph $g$. The atomic labels are then inferred from the generated adjacency matrix using our proposed novel approach.

**Image Feature Extractor Backbone** This component consists of an arbitrary deep neural network that extracts visual features from an input frame. The input to this module is a colour image, $x \in \mathbb{R}^{3 \times W \times H}$ and the output is a feature map $f \in \mathbb{R}^{C \times W_f \times H_f}$. This feature map is reduced to a lower dimension using a $1 \times 1$ convolution operator with R channels. Since the encoder of the transformer network expects a sequence of features, the feature map spatial dimension is collapsed into a single dimension using a flatten operator, resulting in a final feature map $z \in \mathbb{R}^{R \times W_f H_f}$. We compared different backbones (ResNet50(He et al., 2016) and ResNet101(He et al., 2016)) in our experiments.

**Transformer encoder-decoder network** This follows a standard transformer architecture (Vaswani et al., 2017) that consists of a multi-head self-attention and feed-forward networks for the encoder and an additional multi-head cross-attention layer for the decoder.

**Multi-layer perceptron** As shown in the lower right portion of Figure 3, an HTI instance is a tuple containing the human class, interaction class, target class and human and target bounding boxes (x, y, width and height). The human, target and interaction classes consist of binary labels (human/not human, object/not object, and looking/not looking, respectively). The HTI instances are decoded from the output embedding of each HTI query using several MLPs in parallel. We use separate one-layer MLPs with a final softmax layer for each confidence for the human class, target class and interaction class, while separate three-layer MLPs are used for each human and target bounding box.

**Inferring atomic-level gaze communication** We present a novel and effective way to infer atomic-level labels of each person in the scene by exploiting the interesting properties of adjacency matrices. Given an adjacency matrix $A$, each entry $A_{v_i,v_j} = 1$ corresponds to a directed edge from node $v_i$ to node $v_j$. Hence, two persons $(v_i, v_j)$ have *Mutual* gaze behaviours if $A_{v_i,v_j}$ and $A_{v_j,v_i}$ are equal to 1. To determine if a person $v_i$ has a *Shared* attention to a human/object $v_j$ with another person, check if $A_{v_i,v_j}$ is equal to 1 and identify whether the column $v_j$ has more than one entry with a value of 1. If none of these cases is met, the person only has an atomic-level label of *Single*. To illustrate this process, we analyse the frames with adjacency matrices $(A_t, A_{t+1}, A_{t+2})$ in Figure 2. $A_t$ has three nodes $v \in \mathcal{V}$, where $\mathcal{V}$ can be $Person1$, $Person2$ or $Object1$. Looking at $A_t$, $A_{Person1,Person2}$ and $A_{Person2,Person1}$ are both equal to 1, hence the inferred atomic-level gaze communication for both *Person1* and *Person2* are *Mutual*. Looking at $A_{t+1}$, $A_{Person1,Person2}$ and $A_{Person2,Object1}$ are both equal to 1, hence both persons have an atomic-level label of *Single*. Finally, looking at $A_{t+2}$, $A_{Person1,Object1}$ and $A_{Person2,Object1}$ are both 1 (i.e., column $v_{Object1}$ have two entries with a value of 1), hence both persons have *Shared* attention labels.

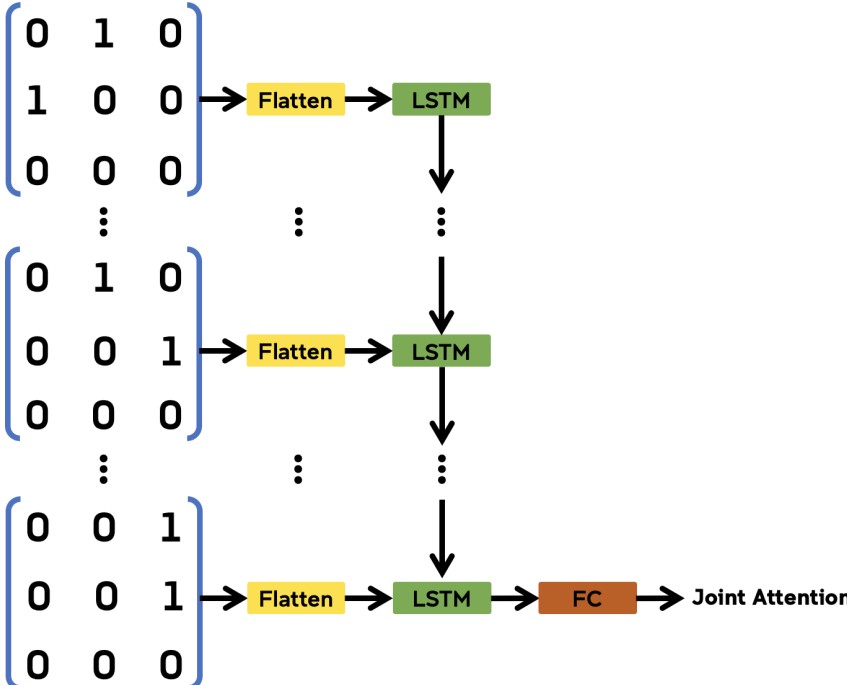

Figure 4: The proposed architecture of the event-level prediction module. It consists of a long short-term memory (LSTM) network and a fully-connected layer for event-level classification. Displayed is a sample input that contains a list of unique adjacency matrices predicted by the atomic-level prediction module and an output that is the final classification of *Joint Attention*.

### 4.3. Event-level prediction module

Our event-level prediction module, as shown in Figure 4, consists of a long-short term memory (LSTM) architecture that takes in a list of unique adjacency matrices predicted by the atomic-level prediction module. Similar to a concept introduced elsewhere (Liu et al., 2021), the proposed network processes the entire video and only uses key frames for prediction. The key frames are defined as the frames where an entry (or entries) of the predicted adjacency matrix has (have) changed. We believe that this simple yet effective approach resembles the behaviour of humans when determining gaze communication events in a video (i.e., humans watching a video keep track of the unique atomic-level labels that happened throughout the video before predicting an event label). In addition, the transition from one atomic-level label to another provides important information about the event-level label that has transpired. Since atomic-level labels can be inferred from the adjacency matrix $A$, keeping track of the unique $As$ across time provides crucial information about the event. Once the key frames have been identified, their corresponding adjacency matrices are flattened and used as input to the LSTM network. Afterwards, the learned features are passed to a fully connected layer for event-level classification.

## 5. Experiments

### 5.1. Dataset

We used the (M)VACATION dataset to train and evaluate our atomic- and event-level prediction modules. Since we predict the atomic-level labels in a parallel and end-to-end manner, the only input to the atomic-level module is an image frame, while the human head and object bounding box labels are used as ground truth to train the model. To prevent overfitting of the atomic-level prediction module, we sample each video every 10 frames and use the sampled frames as inputs to the module. Using the atomic-level ground truth, we generate the adjacency matrices for the entire duration of each event-level label and extract the unique adjacency matrices to train our event-level module.

### 5.2. Implementation Details

Our proposed modules were implemented in PyTorch. The input to the atomic-level module is a normalised RGB image, while the output is a set of HTI instances. To train our model, we used the Hungarian algorithm (Kuhn, 1955) to solve the matching of the predicted HTI instances with the ground truth. After the optimal matching was found, we used the loss function:

$$\mathcal{L}_{loss} = \beta_1 \sum_{c \in h,t,g} \mathcal{L}_{class}^c + \beta_2 \sum_{b \in h,t} \mathcal{L}_{bbox}^b \tag{1}$$

where $\mathcal{L}_{class}^c$ are the standard cross entropy losses between the human, target and gaze interaction and their corresponding ground truth labels. On the other hand, $\mathcal{L}_{bbox}^b$ consists of the weighted sum GIoU loss and L1 loss and is computed for each human and target bounding box.

We used different feature extractor backbones (ResNet50 and ResNet101) pre-trained on the ImageNet database and freeze their batch norm layers. We compared DETR (Carion et al., 2020) and DeformableDETR (Zhu et al., 2020) models pre-trained on COCO for our transformer encoder-decoder and MLP networks. Note that we only used the ResNet50 version of DeformableDETR since the ResNet101 version pre-trained on COCO is not publicly available. We used the default number of encoder and decoder layers, as well as the number of object (HTI in our case) queries, chosen by the original DETR and DeformableDETR models. We set the AdamW optimizer with the following parameters: learning rates of the backbone is 1e-5 and the transformer network is 1e-4; weight decay is set to 1e-4 and applied after 200 epochs. Similar to other DETR-like architectures, our models are trained with a long training schedule (250 epochs).

The event-level prediction module was trained on the ground truth adjacency matrices of the training split of the (M)VACATION dataset. We compute the cross entropy loss between the predicted event-level gaze communication and the ground truth to train the model. For testing, the input of the event-level prediction module is a sequence of unique adjacency matrices predicted by the atomic-level prediction module. The length of the adjacency matrix sequence is set to a value of 5. We set the limit of the maximum number of nodes to 7 based on the (M)VACATION statistics. Smaller adjacency matrices are appended with zeros and shorter sequences are appended with the last unique matrix.

## 5.3. State-of-the-art models

A spatiotemporal graph neural network has been proposed to represent the atomic-level gaze communication behaviours in a given frame (Fan et al., 2019). In addition, an event network was developed to predict the event-level gaze communication behaviours (Fan et al., 2019). While their approach is fundamentally similar, a fair comparison is not possible because of the following reasons: (1) their models were trained and evaluated using the VACATION dataset and (2) their models cannot be re-trained and re-evaluated on the (M)VACATION dataset due to the nature of their open-source implementation (i.e., their source code contains references to files that were not released). Re-implementing their network is out of the scope of this work. Instead, the proposed model was compared to a baseline that predicts the gaze target of a person in a scene (Chong et al., 2020). In particular, a pre-trained model that accepts a human head bounding box, as well as the entire image, was used to predict the probability distribution of the gaze target. Afterwards, the location with the highest probability value was utilised to determine if any human head/object falls within this location and construct an adjacency matrix that can then be used to infer atomic-level labels. This approach is fundamentally similar to the behaviour of the proposed network (i.e., GazeTransformer also predicts the attended target), with the exception that the proposed model performs automatic prediction of human head/object locations instead of using ground truth human head/object locations.

## 5.4. Evaluation Metrics

We use precision ($\mathcal{P}$), F1-score ($\mathcal{F}$) and top-1 average accuracy (Fan et al., 2019) to evaluate both our atomic- and event-level prediction modules. We also report the $\mathcal{P}$, $\mathcal{F}$ and Recall ($\mathcal{R}$) values to demonstrate GazeTransformer's performance in detecting human/object locations. A prediction is considered a true positive if and only if the model predicts a box location that has an intersection-over-union (IOU) greater than 0.5 with the ground truth. To make the number of the atomic- and event-level predictions and ground truth the same and allow for a meaningful comparison, a node without gaze interaction with other nodes is added to the predicted adjacency matrix when ground truth is missed.

## 6. Results

We discuss both the quantitative and qualitative results on atomic-level prediction in Section 6.1 followed by the event-level prediction in Section 6.2.

### 6.1. Atomic-level prediction module

**Quantitative results** GazeTransformer achieved the following human/object localisation performance: ResNet50: $\mathcal{P}$=92.29%, $\mathcal{F}$=83.58%, $\mathcal{R}$=76.38%, while ResNet101: $\mathcal{P}$=90.23%, $\mathcal{F}$=83.65%, $\mathcal{R}$=77.97%, suggesting that the models reported low false positives but moderate false negatives. This means that most of the time, no model generated bounding box predictions that do not contain any human head or object. However, there were times when the models failed to generate bounding boxes that should be there. In the next section, the reasons that contributed to this performance are explored, specifically the missed predictions on small objects.

| | Atomic-level Gaze Communication (Precision $\mathcal{P}$, F1-score $\mathcal{F}$ & Average Accuracy) | | | | | | |
|---|---|---|---|---|---|---|---|
| | Single | | Mutual | | Share | | Average Accuracy |
| | $\mathcal{P}(\%)\uparrow$ | $\mathcal{F}(\%)\uparrow$ | $\mathcal{P}(\%)\uparrow$ | $\mathcal{F}(\%)\uparrow$ | $\mathcal{P}(\%)\uparrow$ | $\mathcal{F}(\%)\uparrow$ | top-1 $(\%)\uparrow$ |
| Ours* | 79.48 | **86.28** | **75.57** | 66.38 | 94.81 | 60.06 | **88.40** |
| Ours† | **79.98** | 85.17 | 72.78 | **68.12** | 83.10 | 60.24 | 87.88 |
| Ours‡ | 77.93 | 85.24 | 70.59 | 58.85 | **95.17** | **64.16** | 87.54 |
| (Chong et al., 2020) | 70.05 | 68.61 | 43.42 | 58.21 | 64.31 | 33.34 | 78.92 |

Table 2: Quantitative performance of different models on (M)VACATION dataset for atomic-level gaze communication prediction. Model used: *DETR with ResNet50, †DETR with ResNet101, ‡DeformableDETR with ResNet50.

The atomic-level classification results are reported in Table 2. All GazeTransformer models achieved promising precision ($\mathcal{P}$) values on all atomic-level classes with the highest $\mathcal{P}$ value of 95.17% on *Share* class, much higher than the baseline model (Chong et al., 2020). Similarly, the GazeTransformer models reported higher F1-scores ($\mathcal{F}$) than the baseline model. Overall, the GazeTransformer models achieved a similar high average accuracy of around 88%. In comparison, the baseline model achieved a lower performance across all the atomic-level gaze communication behaviours. In particular, the baseline model resulted in a 10% lower precision ($\mathcal{P}$) value of 70.05% and a 20% lower F1-score ($\mathcal{F}$) of 68.61% on the *Single* class. In addition, it resulted in around 30% lower precision ($\mathcal{P}$) value of 43.42% and a 10% lower F1-score ($\mathcal{F}$) of 58.21% on the *Mutual* category. The greatest difference in performance was on the *Share* class where the baseline model reported a 30% lower precision ($\mathcal{P}$) value of 64.31% and around 30% lower F1-score ($\mathcal{F}$) of 33.34%. Overall, the baseline model had a 10% lower average accuracy of 78.92%. All variants of GazeTransformer consistently outperformed the baseline model (Chong et al., 2020). In the next section, we will focus on the GazeTransformer with ResNet50 and DETR-like architecture.

**Qualitative results** As shown in the first two rows of Figure 5, the proposed Gaze-Transformer correctly predicted human head and object locations (in coloured solid rectangles) that are close to the ground truth (in red dotted rectangles). Directed arrows are added to show the predicted attended targets of all the detected humans. In addition, it can correctly infer *Single*, *Mutual* and *Share* atomic behaviours in scenarios where there are exactly two persons (Columns 1-3) or even three persons (Columns 4-5).

GazeTransformer also predicted labels that were different from the ground truth (for the succeeding discussion, refer to the last two rows of Figure 5). Column 1 shows that our model predictions were *Mutual*, while the ground truth was *Single*. Here, the subtle cue of eye gaze direction results in a drastically different atomic-level label. This illustrates why our model achieved lower $\mathcal{P}$ and $\mathcal{F}$ values for the *Mutual* label. We found instances where we believe that the ground truth (*Mutual*) was incorrect (column 2). Our model was penalised for predicting the correct *Single* label, effectively lowering our model's *Single* and *Mutual* performance. As shown in columns 3-4, our model failed to correctly predict the locations of small objects, resulting in lower $\mathcal{F}$ on *Share* label. This is also reflected in the lower $\mathcal{R}$ value ($\sim$ 76%) of GazeTransformer's localisation performance. Finally, column 5 shows ambiguous cases that are too difficult even for humans to identify.

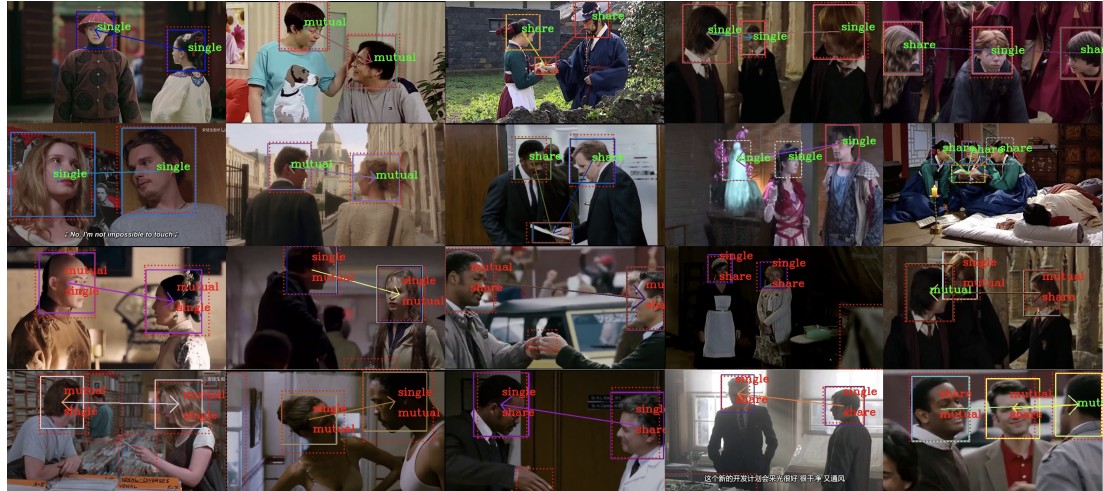

Figure 5: Atomic-level prediction results. The first two rows show frames with correctly classified labels (in green), while the last two rows show frames with incorrectly classified labels (in red, above the ground truth). The dotted rectangles are ground truth labels, while solid rectangles are model predictions.

## 6.2. Event-level prediction module

The event-level classification results are shown in Table 3. Our temporal module combined with different GazeTransformers has promising $\mathcal{P}$ and $\mathcal{F}$ values for *Non-Communicative*, *Mutual Gaze* and *Joint Attention*, but low performance for *Gaze Aversion* and *Gaze Following*. Overall, our temporal modules have a similar high average accuracy of around 85%. To eliminate any compounded errors caused by GazeTransformer and illustrate the effectiveness of our temporal module alone, we fed the latter with the ground truth adjacency matrices and achieved significantly higher performance across all metrics on all event labels, highlighting that our temporal module is working as intended. On the other hand, the baseline model was not able to predict *Gaze Aversion* and *Gaze Following* classes and a low performance on *Joint Attention*, resulting in a much lower average accuracy of 22.90%.

| | Event-level Gaze Communication (Precision $\mathcal{P}$, F1-score $\mathcal{F}$ & Average Accuracy) | | | | | | | | | | |
|---|---|---|---|---|---|---|---|---|---|---|---|
| | Non-Comm. | | Mutual Gaze | | Gaze Aversion | | Gaze Following | | Joint Attention | | Avg. Acc. |
| | $\mathcal{P}(\%)\uparrow$ | $\mathcal{F}(\%)\uparrow$ | $\mathcal{P}(\%)\uparrow$ | $\mathcal{F}(\%)\uparrow$ | $\mathcal{P}(\%)\uparrow$ | $\mathcal{F}(\%)\uparrow$ | $\mathcal{P}(\%)\uparrow$ | $\mathcal{F}(\%)\uparrow$ | $\mathcal{P}(\%)\uparrow$ | $\mathcal{F}(\%)\uparrow$ | top-1 (%) $\uparrow$ |
| Ours* | 61.76 | **62.69** | **68.42** | 65.00 | **25.00** | 33.33 | 20.00 | 18.00 | 43.75 | 46.67 | 85.03 |
| Ours† | 60.60 | 61.54 | 57.14 | 63.15 | 25.00 | 33.33 | **25.00** | 25.00 | **71.42** | **58.82** | **85.23** |
| Ours‡ | 58.82 | 57.97 | 57.14 | 58.54 | 25.00 | 33.33 | 20.00 | 18.00 | 50.00 | 53.85 | 83.71 |
| (Chong et al., 2020) | **71.43** | 31.25 | 23.81 | **71.43** | 0 | 0 | 0 | 0 | 37.5 | 33.33 | 22.90 |
| Ours†† | **74.60** | **82.46** | **89.66** | **85.25** | **66.67** | **66.67** | **66.67** | **72.73** | **78.57** | **57.89** | **91.90** |

Table 3: Quantitative evaluation results of different models on (M)VACATION dataset for event-level gaze communication prediction. Model used: *DETR with ResNet50, †DETR with ResNet101, ‡DeformableDETR with ResNet50, ††The input is the ground-truth adjacency matrices instead of the atomic-level predictions.

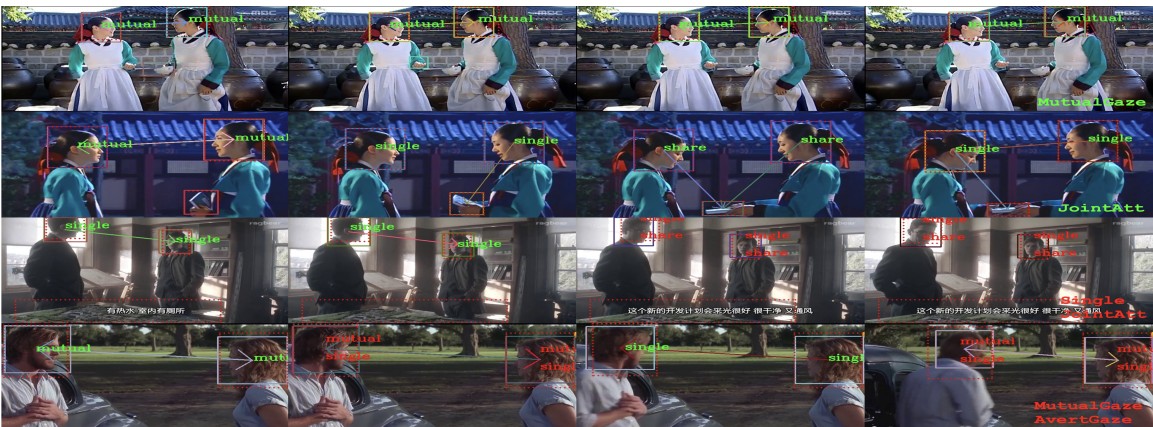

Figure 6: Event-level prediction results. The first two rows show correctly classified atomic and event labels (in green), while the last two rows show incorrectly classified atomic and event labels (in red, above the ground truth).

Figure 6 illustrates the qualitative results. A closer examination of Rows 1 and 2 reveals that all the frames depict accurate atomic-level predictions, resulting in correct event-level predictions. On the other hand, the event-level predictions in Rows 3 and 4 are incorrect because GazeTransformer either failed to identify an object in the scene (as observed in Row 3) or incorrectly predicted the atomic-level labels of one or two persons in the scene (as depicted in Row 4). This affirms the heavy reliance of the temporal module on the Gaze-Transformer. More examples of atomic- and event-level predictions on the (M)VACATION dataset are currently available in the GitHub repository linked in the Abstract.

## 7. Conclusion

We have presented a two-stage approach for the temporal understanding of gaze communication. Compared to previous approaches, our first stage does not require human head/object bounding box locations. Instead, our module predicts these locations, attended targets and their corresponding gaze relationships in parallel. Unlike previous end-to-end models that can only predict attended gaze targets or identify *Mutual* gaze instances, our model can infer *Single*, *Mutual* and *Share* gaze behaviours. Afterwards, a temporal model uses the predicted atomic-level labels to identify event-level gaze communication. Both models show promising results on the (M)VACATION dataset. Despite the encouraging results, our proposed two-stage framework has some notable limitations. First, our atomic-level module has a lower localisation performance on small objects, as shown by the failure cases in Figure 5, resulting in lower atomic-level classification performance. Second, our event-level module is heavily dependent on our atomic-level module since both modules are in cascade. This is substantiated by the increase in performance of the event-level module when the ground truth adjacency matrices were used as input. Despite this promising result, the event-level results may still be sub-optimal, which is caused by the disjointed training of both modules (due to limited data). Nevertheless, our approach offers an end-to-end solution for atomic-level prediction combined with a temporal module for event-level prediction.

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
