# OpenReview forum: "Temporal Understanding of Gaze Communication with GazeTransformer"
_NeurIPS.cc/2023/Workshop/Gaze_Meets_ML — Gaze Meets ML 2023 Poster_

### Official Review · Reviewer_om7c · 2023-10-22
**A novel end-to-end approach for temporal understanding of gaze communication**

**Rating:** 6
**Confidence:** 4

**Review:**

The authors introduce an end-to-end approach to address the challenge of gaze communication. They tackle this problem by operating at both the atomic level, using a transformer-based architecture, and the event level, employing a temporal model. Since the closest related work is Fan et al. (2019), the authors detail the modifications they made in their approach, which present the problem formulation from a different angle.

As the main contribution, the authors present a new dataset, although the changes made to it could be considered not significant, but mainly simplifications of the original dataset. While the authors provide a reasonable explanation for potential issues with the baseline dataset, relying on a single example (depicted in Figure 1) is not sufficient to support their decision to modify the dataset to such a significant extent. It is believed that additional examples are necessary to justify the removal of certain classes.

One notable concern is the absence of a thorough comparison with other state-of-the-art methods that are closest in approach to the proposed work. Although the authors acknowledge this concern in the article, testing on the baseline dataset, in this case, could be essential for a fair comparison with the state-of-the-art, particularly with a similar approach like Fan et al. (2019).

---

### Official Review · Reviewer_t311 · 2023-10-23
**Temporal Understanding of Gaze Communication with GazeTransformer**

**Rating:** 6
**Confidence:** 4

**Review:**

Strengths:
(+) Addresses relevant problem by proposing an end-to-end solution instead of a decoupled or two-branch solution.

Weaknesses:
(-) Clarity: Presentation clarity through figure visualizations of model outputs could be improved
(-) Evaluation: Single baseline adapted to task as opposed to current state-of-the-art comparison

This paper presents an end-to-end model for understanding gaze communication within video content. Gaze communication refers to humans looking independently, at each other, or at common objects. The task is highly relevant for computer vision, gaze estimation, and collaboration. The paper is reasonably well organized, though some of the links between text and figure visualizations could be improved to assist the reader in following the authors' takeaways. The strongest contribution of the paper is an end-to-end two-stage model to address the research task.

The significance of an end-to-end model for this task is strong, and the evaluation split of atomic and event-level analysis helps the paper by decoupling atomic-level model performance from the overall understanding of how well the model is performing. Standard transformers and modeling approaches from the field are applied, but a novel re-labeling is introduced to the evaluation dataset as well. The evaluation has sound depth, covering atomic-level and event-level performance, as well as a few feature extraction backbones and consideration of ground truth adjacency matrices. The findings are encouraging and seem to advance knowledge for this task.

The weak points of this paper center around clarity of presentation and limitations in the breadth of the evaluation. Specifically, following text to understand what is happening with different inferences in different scenarios was difficult at times, due in part to the opaqueness of the text covering the content as well as a lack of context over what is happening in each set of frames (Figure 1). Saying what is happening in the scene, or moving the text labels around may alleviate this issue. Second, the authors acknowledge that re-implementing the state-of-the-art is not possible due to missing files. This is a valid point. However, the current evaluation adapts only one existing baseline (Chong 2020). If my interpretation is correct, this could be addressed by applying the pre-trained model from Fan et. al 2019 on VACATION and only considering the common single/mutual/share atomic instances between VACATION and (M)VACATION due to the effect of re-labeling. Such a comparison would give a better understanding of the benefits of the proposed model.

Overall, while there are limitations in the evaluation, I believe the end-to-end nature of the system and findings with regard to two-stage classification seem novel and useful for the field. Clarity in presentation issues are small enough to be addressed by the authors prior to the presentation. I scored the paper as a 6 as a result.

Other Comments:
-Fig. 1, it is difficult to interpret which scenario is referenced on page 5 as well as it is hard to see the actual frame content due to opaqueness of text and arrows over the frame

-The text on page 6 suggests that Fig. 1 shows consecutive frames. It does become clear that each row is from same video, but how much time has passed and what is guiding the gaze behavior makes it somewhat difficult to grasp what is actually going on here and the resulting inferences on social gaze behaviors

-Fig. 5, hard to grasp which of two red labels is predicted and ground truth.

---

### Official Review · Reviewer_3Nqm · 2023-10-24
**Relevant paper for the workshop, well formulated problem, and well written procedure. The only lacking part is the evaluation of the results once a single dataset and baseline.**

**Rating:** 6
**Confidence:** 4

**Review:**

# Review for "Temporal Understanding of Gaze Communication with GazeTransformer"

## Quality
The quality of the paper, approach, and methodology is above average. The quality of the evaluation methods is lacking.

## Originality
The application and methodology are novel as far as I am aware.

## Significance and Impact
This work is a significant first step, with a limited impact due to limited evaluation methodologies. The broader significance of this work is as a proof of concept in the application of Transformer blocks as a low-resource atomic gaze predictor.

## Clarity and Writing
The paper is well structured, the motivation and approach are clear  and well written. The description of both modules is easy for follow and should be relatively uncomplicated for any future re-implementations.

## Pros
- The paper is well-written and easy to understand.
- Both atomic and event-level modules are clearly described and reproducible with the github source code.
- Human head location is also predicted by this approach unifying the gaze communication prediction with the extraction of the target head.
- The paper introduces a modified version of a popular dataset, that is more balanced for labels and logically accurate.
- The joint attention mechanism employed for the event level prediction module is well documented in literature.
- Single framework for the Inference of three types of gaze communications is useful as a proof-of-concept, albeit with limited use outlined.
- The pre-trained models used are publicly available, thus making the approach independently re-implementable. This is important for research that leverages pre-trained networks.


## Cons
- The process by which the removed class labels from VACATION dataset are assigned to the three labels of (M) VACATION is not clear. I understand paper space limitations but a figure or concrete formulae for labelling in the modified dataset are crucial bits of information.
- Some design choices are not well-explained. For example, the reasons for using transformer for atomic predictions while LSTMs for event-level predictions are not clear. Cascading nature of the two modules intensifies the need for a proper ablation with different RNN architectures in each module.
- Lacking thorough evaluation in terms of dataset ( a single dataset) and most importantly, baselines (only 1 real baseline). Results outline 3 baselines but two of them are basically ablations/versions of the proposed approach. Future versions would benefit from more baselines, and moving the comparison of authors' own approaches to a separate ablation section.

---

### Meta-Review · Area_Chair_hx42 · 2023-10-26

**Recommendation:** Accept (Poster)
**Confidence:** 4

**Metareview:**

This paper proposed a GazeTransformer that infers atomic-level behaviors in a given frame and a temporal module that predicts event-level behaviors in a video using the inferred atomic-level behaviors. Reviewers commended the significance of the work. However, they pointed out some paper weaknesses, which some can be addressed in the camera-ready version.

---

### Decision · Program_Chairs · 2023-10-26

Accept (Poster)